# Longitudinal Training and Workload Assessment in Young Friesian Stallions in Relation to Fitness: Part 1

**DOI:** 10.3390/ani13040689

**Published:** 2023-02-16

**Authors:** Esther Siegers, Emma van Wijk, Jan van den Broek, Marianne Sloet van Oldruitenborgh-Oosterbaan, Carolien Munsters

**Affiliations:** 1Department of Clinical Sciences, Faculty of Veterinary Medicine, Utrecht University, Yalelaan 114, 3584 CM Utrecht, The Netherlands; 2Equine Integration, Groenstraat 2C, 5528 NS Hoogeloon, The Netherlands

**Keywords:** standardized exercise test, performance, horses, Friesian breed, workload, training

## Abstract

**Simple Summary:**

Young Friesian stallions have to complete a 10-week training program (70-day test) to become studbook approved breeding stallions. The effect of this training program, consisting of dressage and driving training, on fitness progress was evaluated in this study. Data were collected in the 9 weeks preceding the training program, and during the 70-day test. Duration and time spent at walk, trot and canter for all training sessions were measured in the participating 16 stallions. Each stallion had to perform three ridden standardized exercise tests in week 1, 6 and 10 of the 70-day test, during which heart rate and plasma lactate concentration were measured. Horses were trained for a longer duration in the 70-day test compared to the preceding weeks. Heart rate and plasma lactate concentration of the horses increased in the second and third exercise test compared to the first exercise test, indicating that the stallions were more fatigued during the second and third exercise tests. These results indicate that the fitness of Friesian stallions decreased during 70-day test, suggesting overtraining. The 10-week training program needs to be adjusted to optimize training effects and welfare of the horses.

**Abstract:**

Friesian stallions have to complete a 10-week studbook approval test (70-day test) consisting of dressage and driving training for acceptance as a breeding stallion. Training response of the horses to this approval program was evaluated. External workload (EW) of 16 stallions (3.2 ± 0.4 years old) was registered during the 9 weeks preceding the 70-day test (preparation period) and during 70-day test. Ridden indoor standardized exercise tests (SET), measuring heart rate (HR) and plasma lactate concentration (LA) at walk, trot and canter, were performed in week 1 (SET-I), 6 (SET-II) and 10 (SET-III) of the 70-day test. Linear mixed effect models were used to analyze HR and LA related to SETs and EW related to the phase of the study, using Akaike’s information criterion for model reduction. EW was higher in the 70-day test compared to the preparation period, with longer duration of training sessions. Compared to SET-I, horses showed increased HR and LA after canter in SET-II and SET-III, indicating that they were more fatigued. The fact that the fitness of these Friesian stallions decreased during the 70-day test suggests overtraining. To optimize training effects and welfare of these horses, the workload of the training program needs to be adjusted.

## 1. Introduction

Training evaluation of horses is of interest to riders, trainers, veterinarians and other equine professionals. Adequate training prepares the equine athlete for the physical demands of a task or performance by inducing a multitude of physical adaptations [1]. However, unsuitably high or low training loads may result in an increased injury risk or poor performance [2,3]. In an appropriate training protocol, an intense training session should be followed by recovery time since intense exercise leads to disruption of homeostasis in cells and organs. In the recovery phase, cellular repair occurs and supercompensation follows. Over time, this leads to increased performance. When athletes take insufficient recovery time repeatedly, they are at risk for overreaching or overtraining [2,4,5,6]. Daily monitoring of training load is common in human sports [7], allowing for optimal and individualized training programs. However, quantification of workload in Warmblood sport horses is not yet common. Knowledge about safe and effective training programs will contribute to equine welfare, optimal performance and injury prevention. The current developments in equine wearables, such as heart rate sensors with GPS and/or accelerometers, make it more and more accessible to horse trainers to quantify workload on a daily basis, identify performance indicators and investigate factors related to injury occurrence [8]. 

Training load, or workload, can be divided into external load and internal load. External load is an objectively measurable external stimulus, such as speed, running distance, number of sprints or number of jumps. Internal load is the physical and sometimes behavioral response to external workload, such as heart rate (HR), plasma lactate concentration (LA) or perception of the exercise by the athlete [3,7,9,10,11]. The internal load gives information on how intense the external load for the individual is, and it is the stimulus for physiological adaptation to training [12]. The combination of external and internal workload gives information about the fitness level of the athlete. Detailed data about training programs and workload in different equestrian disciplines and their impact on fitness are currently sparce. In thoroughbred racehorses and standardbreds, several studies investigated the physiologic responses to training programs [1,2,3,4,5,6,7,8,9,10,11,12,13,14,15]. However, in daily practice, training programs of sport horses currently often lack an evidence base [16,17]. Evaluation of training programs and/or training load in horses is often acquired by general estimation using surveys or questionnaires. These methods have their limitations and are subjective to several forms of data bias [17,18,19,20,21]. More accurate information about horses’ gait, movement, duration of training, speed, distance covered and individual impact on the horse can be acquired by using GPS, accelerometers and HR monitors [22,23,24,25,26,27]. With the workload data, indices can be calculated such as the cumulative workload index, the acute to chronic workload ratio (ACWR) and high-speed exercise (HSE) distance [24,28,29]. These measures for workload are used to evaluate training programs and to identify relationships with the occurrence of injuries [24,29,30]. Most of these measures use speed and distance as the most important, or only determinants for workload quantification. It is, however, difficult to use these measures in Warmblood sport horse disciplines such as dressage, where speed and distance are not the crucial determinants of the intensity of the exercise [31,32]. Therefore, other measures need to be used such as the intensity of exercise by using duration and HR calculations [8,33], or iTRIMP (individualized training impulse) [24,34]. 

Standardized exercise testing (SET) is a proven method to assess aerobic capacity in horses [32,35,36]. A field exercise test typically exists of several incremental bouts of exercise during which HR (bpm), LA (mmol/L) and velocity (m/s) are measured [35]. Using repeated SETs during a period of time may identify individual differences, evaluate fitness levels and assess training responses of horses to a certain training program [32,35,36,37]. Higher HR or LA and/or slower recovery HR are associated with poor performance or possible underlying (subclinical) disease [35,37,38]. Some studies even show a relation between fitness indices and the incidence of injuries [24,32]. For Warmblood horses, testing under field conditions is the most representative for the environment in which these horses have to perform. The design of the test should be adapted to meet the specific demands asked of the horse in competition or training [32,39]. Dressage demands different technical skills of a horse, compared to show jumping or eventing [32,40,41]. 

Friesian horses in the Netherlands are mainly used for recreational riding, dressage or driving. The breed has a strictly closed studbook, which has a low genetic diversity and thus a high inbreeding coefficient (25.5%) [42]. Inbreeding is thought to be the cause of several (suspected) genetic disorders. Conditions such as dwarfism and hydrocephalus, distichiasis, retained placenta, insect bite hypersensitivity, megaesophagus, chronic progressive lymphangitis and ruptures of the aortic arch have a high prevalence in the Friesian breed [42,43,44,45,46]. Young Friesian horses seem to have a higher energy expenditure in trot and canter. They show higher HR and LA results in comparison to Warmblood horses with comparable backgrounds, trotting and cantering at the same speed [47,48,49], indicating that the transition to anaerobic metabolism starts at lower exercise speeds in Friesian horses. De Meeûs d’Argenteuil et al. performed a study in untrained Friesian horses undergoing an 8-week low intensity training protocol on a treadmill or in an aquatrainer. The increase in muscle diameter was the most after 4 weeks of training. Training intensity did not change after 4 weeks of training and muscle diameters decreased again. These results suggest that in an appropriate training protocol, a slight increase in workload should be made after 4 weeks of training. Shifts in muscle fiber type and metabolic profiles occurred after the training protocols, with different muscles showing different changes [50]. 

Friesian stallions, selected by the Royal Friesian Horse Studbook (KFPS) to be considered as future breeding stallions, are brought together annually at a training facility. They are trained and educated for 70 consecutive days in a studbook approval test (70-day test) to examine their ability for dressage and driving performance as well as their characters and overall health. In the first six weeks, the young stallions are mainly trained under saddle. In the following four weeks, the horses are also trained in driving [51]. In general, the welfare of sport horses is more and more under the scrutiny of the public. Evaluating current practices and optimizing welfare is essential. In that light, the KFPS wanted to evaluate the current training program of the young Friesian studbook stallions during the 70-day test.

The aim of the study was to evaluate the workload of the current (2020) training program used during the 70-day test and its effect on the participating young Friesian stallions.

## 2. Materials and Methods

### 2.1. Study Design

The study consisted of two parts: (I) the preparation period before the 70-day test and (II) the actual 70-day test (see Figure 1). The 70-day test is the final exam for Friesian stallions to become an approved studbook stallion at the KFPS. In the period before of this 70-day test (= preparation period) a large group of stallions (*n* = 56) is evaluated by official studbook judges on three specific selection days. In the end, only 16 stallions are selected by the judges to participate at the 70-day test as their last test to become an approved studbook stallion. Inclusion criteria for horses to participate in the first part of the study were as follows: horses being selected by the official studbook judges of the Friesian studbook (KFPS) to participate at the three selection days; owner consent to measure HR and LA during the selection days. Inclusion criterion for the second part of the study, the 70-day test, was being selected by the KFPS judges to participate in the 70-day test. 

#### 2.1.1. Preparation Period and Selection Days

All potential studbook stallions had to participate in three studbook selection days in the months prior to the 70-day test. Due to COVID-19 restrictions, data collection was not possible on the first selection day of the KFPS and thus data collection for the present study started on the second selection day (referred to as selection day-A). In total, the preparation period consisted of a 9-week period, including two selection days: day-A and day-B. The preparation period started on selection day-A and continued until the start of the 70-day test. Selection day-B was held five weeks after selection day-A. During the preparation period, horses were trained at home by their individual rider or trainer in their normal environment. On the KFPS selection days, horses performed a ridden test at walk, trot and canter in front of the official judges of the Friesian studbook. Based on their total performance the judges decided which stallions were allowed to enter the 70-days test. The studbook judges based their decision on subjective selection criteria such as exterior traits, gaits and character as well as genetic information and health assessments such as sperm quality, left laryngeal hemiplegia and osteochondrosis dissecans [51].

#### 2.1.2. 70-Day Test

During the 70-day test, the training program was determined by the head trainer of the studbook (KFPS). During the 70-day test, the horses’ characters, dressage and driving abilities and health were evaluated. Horses were trained by 4 experienced riders and 2 experienced drivers. Horses were trained by the same rider as much as possible. The stallion’s performance, behavior, stable manners and the occurrence of any veterinary problems were documented. Studbook judges assessed the horses twice for their studbook suitability in week 6 and week 10. Stallions could be eliminated from the 70-day test based on the judges score, or as a result of an injury or other illness assessed by a veterinarian. During the 70-day test, three standardized exercise tests (SETs) were performed in week 1 (SET-I), week 6 (SET-II) and week 10 (SET-III).

#### 2.1.3. Horses

Data were collected from privately owned young Friesian stallions. In the first part of the study, the preparation period, 38 stallions at the age of 3 (*n* = 31) or 4 (*n* = 7) years were allowed to participate by their owners. In the second part of the study, the 70-day test, 16 stallions at the age of 3 (*n* = 13) or 4 (*n* = 3) were selected by the KFPS to participate. During the entire study (preparation period and 70-day test), horses were kept at individual stables and were fed an individual diet consisting of roughage and pellets. Water was provided at libitum. The Animal Ethics Committee of Utrecht University approved all research procedures (reference number 5204-1-04, approval date 24 June 2020). Written owners’ informed consent was also obtained. 

### 2.2. Equipment

During the selection days, HR measurements were performed using a HR sensor (Polar V800^®^ Polar Electro Oy, Kempele, Finland and Equisense motion S © Equisense, Bidart, France). During the SETs in the 70-day test, HR recording were performed using the Polar devices. At the selection days and during the 70-day test, duration of the exercise and the time spend in each gait was measured using a 9-axis motion sensor (IMU, Equisense Motion S ©, Equisense, Bidart, France) [52,53] that was placed around the girth, exactly in the middle between the front legs of the horses. Blood samples were collected by venipuncture of the left or right jugular vein using a sterile 2 mL syringe and a 23-gauge needle to determine plasma lactate concentrations (LA, mmol/L; Lactate Pro 2^®^ (Arkray Inc., Kyoto, Japan)). Ambient temperature (°C) and relative humidity (RH, %) were measured during the SETs in the 70-day test using a heat stress wet bulb globe temperature (WBGT) device (Extech instruments HT30, Nashua, NH, USA).

### 2.3. Data Collection

#### 2.3.1. Preparation Period

During the preparation period, the trainer of each stallion filled in a weekly questionnaire to collect their training protocol. Data collected were the type of training (lunging, dressage, hacking, driving, other), frequency of training sessions and duration of a training session. During selection day-A and B, horses were equipped with HR and IMU sensors. HR and LA measurements were performed, and the duration of riding session was obtained. Warming up was conducted according to individual riders’ choice. During the test, horses had to show walk (~1.5 min), trot (~2 min) and canter (~1.5 min) on both hands (total ~10 min) in front of the judges. LA was determined after 60–90 s after completion of the test. From horses with a LA > 2.0 mmol/L, a second blood sample was taken after 10 min of walk. 

#### 2.3.2. 70-Day Test

In the 70-day test horses were equipped with an IMU sensor during all training activities to measure the duration of the training and time spend in each gait. In addition, type of training was registered (riding, lunging, driving). If IMU (duration) data from a training session were missing, measures were estimated using the average duration from training sessions of the same type of that specific horse in the same period. 

#### 2.3.3. Standardized Exercise Tests 

The SETs consisted of three incremental steps (see Table 1). During SETs, duration of the exercise, time spend per gait, HR and LA were measured. Blood samples to determine LA were taken after 30–60 s after each step of the SET, and after 10 min of recovery at walk. From data obtained during the SET, calculations were made to determine the relationships of LA to HR as an exponential regression curve. The HR_LA2_ and HR_LA4_ (HR at LA of 2 and 4 mmol/L) were determined by interpolation. Recovery HR after 10 min were determined after the last exercise step, from which LA reduction was calculated (maximum LA–recovery LA). The protocol for SET-I, II and III was identical for all horses and is shown in Table 1. 

### 2.4. Data Analysis

External workload was determined as duration of each training session and time spent at a certain gait in minutes (walk, trot and canter). Before processing, all HR data were visually checked for artefacts. If many artefacts were present (>5% of the measurement), data were not used for analysis. For HR analysis in the different gaits, the mean HR of the last 60 s in each gait and after 5 and 10 min of recovery were used. For total mean HR of an exercise, all HR data were used from start of exercise until after recovery. Statistical analysis was performed using R-studio^®^ (Boston, MA, USA). 

#### 2.4.1. Number of Training Sessions

The number of training sessions (total and dressage) were analyzed based on the number of training sessions per week, with a maximum of 7 training sessions in 1 week. The number of training sessions were modelled with a logistic regression (grouped binominal data), with random horse intercepts and with period in the study as the predictor variable. 

#### 2.4.2. Selection Days, Workload and SET Data

For the selection days, external workload and SET data, linear mixed effect models were used. Normal probability plots of the residuals were made, and all data proved to be normally distributed. For selection days data (duration, HR, LA), horse ID was a random effect, and age of the horse, whether the horse was selected for 70-day test or not and selection day-A or day-B were fixed effects. For external workload, horse ID was a random effect, and period in the study and age of the horse were fixed effects. For the SET data (HR, LA, LA reduction, HR_LA2_ and HR_LA4_), horse ID was a random effect, and gait (trot, canter-1, canter-2 and recovery after 10 min), SET (I, II or III), age, rider and gait–SET interaction were fixed effects. Akaike’s information criterion (AIC) was used for model reduction. For important effects in the final model 95% confidence intervals (95% CI) were calculated and presented as estimate and 95% CI. All data are presented as mean ± SD.

## 3. Results 

### 3.1. Selection Days

During the preparation period, 36 stallions met the inclusion criteria to be measured on selection day-A and 32 stallions on selection day-B. Of these horses, thirty stallions were measured on both days. From the 16 horses accepted for the 70-day test, 11 stallions were measured on both selection day-A and selection day-B. On selection day-A, HR and duration data for three horses were missing. On selection day-B, HR and duration data for five horses were missing.

Duration, HR and LA results from the selection days are presented in Table 2. Comparing selection day-A and day-B, data did not show differences in HR or LA. When comparing horses accepted for the 70-day test to horses not accepted to the 70-day test, data did not show differences either. Total duration (min) of the exercise was shorter in horses participating in the 70-day test (estimate −4.1; 95% CI −8.0, 0.1) compared to horses not accepted for the 70-day test. Overall, total duration of the exercise (estimate −3.7; 95% CI −7.0, −0.2) and the warm-up period (estimate −3.6; 95% CI −6.4, −0.7) were shorter on selection day-B compared to day-A. 

### 3.2. The 70-Day Test

In total, 16 horses were accepted for the 70-day test, of which 4 stallions completed the 70-day test and were approved as a Friesian studbook stallion; 12 horses were eliminated during the 70-day studbook approval test, 7 horses were eliminated due to judge decisions and 5 horses due to an injury. One stallion entered the 70-day test two weeks after the start of the 70-day test because of locomotory issues; this horse finished the rest of the 70-day test. 

#### 3.2.1. External Workload 

In the preparation period, training data from all 16 stallions that entered the 70-day test were obtained by the questionnaire. During the 70-day test a total of 491 training sessions were analyzed. IMU data from 59 (12%) training sessions were lost due to technical issues and replaced by a calculated average from comparable training sessions for that individual stallion. Duration per training session and training frequency were used to evaluate workload. According to the AIC, the fixed factors period of the study and age of the horse were essential factors which affected the duration of the training sessions of the horse. Fixed factors period of the study and dressage training were essential factors for training frequency. In the preparation period, horses were trained 4.5 ± 0.9 times a week with a mean duration of 122 ± 25 min per week per horse. No difference was shown in training frequency between the preparation period and the 70-day test. In the weeks between SET-I and SET-II, horses were trained longer (minutes/week) compared to the preparation period (estimate 22.4; 95% CI 15.7, 29.0) (Table 3). According to the AIC, total training duration data did not show differences in the weeks between SET-II and SET-III compared to the preparation period. Age of horses influenced their training time significantly; average duration of training sessions per week in 4-year-old horses was shorter (estimate −16.8; 95% CI −29.4, −4.2) compared to 3-year-old horses. The frequency (training sessions per week) of the type of training was similar between the preparation period and the period between SET-1 and SET-II, but horses had less dressage training sessions in the period between SET-II and SET-III (estimate −0.4; 95% CI −0.9, −0.2). For driving and lunging training sessions, there were not enough data in the preparation period and the period between SET-I and SET-II for statistical analysis. 

#### 3.2.2. Standardised Exercise Tests (SETs)

During the 70-day test, 15, 8 and 4 horses participated in SET-I, II and III respectively. Relative humidity and ambient temperature on the days of the SETs were 16.8 ± 3.0 °C and 69.9 ± 10.3% RH on the day of SET-I, 9.4 ± 0.4 °C and 81.4 ± 1.4% RH on the day of SET-II and 10.0 ± 0.5 °C and 71.0 ± 4.1% RH on the day of on SET-III. 

Mean ± SD HR and LA results from SET-I, II and III are shown in Figure 2 and Figure 3. According to the AIC, rider and age of the stallion did not have effects on HR or LA results in SETs. Heart rate was affected by gait and by SET, but not by gait–SET interaction. Compared to canter-1, HR (bpm) was higher in canter-2 (estimate 8; 95% CI 3, 13) and lower in trot (estimate −35; 95% CI −41, −29) and lower after 10 min recovery (estimate −75; 95% CI −80, −70). Compared to SET-I, HR was higher in SET-II (estimate 11; 95% CI 6, 16) and SET-III (estimate 17; 95% CI 11, 24). HR also increased between SET-II and SET-III (estimate 7; 95% CI 1, 13).

LA was affected by gait, SET and gait–SET interaction. Overall, LA (mmol/L) was lower in trot compared to canter-1 (estimate −2.0; 95% CI −2.5, −1.4) and LA decreased after 10 min of recovery (estimate −1.8; 95% CI −2.4, −1.2) compared to canter-1. Compared to SET-I, LA increased in canter-2 in SET-II (estimate 0.8; 95% CI 0.1, 1.4). In addition, compared to SET-I, LA increased in canter-1 (estimate 2.1; 95% CI 1.2, 3.0) and canter-2 (estimate 2.8; 95% CI 1.9, 3.7) in SET-III. When comparing SET-II with SET-III, it was shown that LA increased in canter-1 of SET-III (estimate 2.2; 95% CI 1.0, 3.4) and canter-2 of SET-III (estimate 2.9; 95% CI 1.7, 4.1). Overall LA after 10 minutes of recovery was not different compared to LA after trot. Reduction in LA from maximum LA to LA after 10 minutes of recovery was 66%, 68% and 78% in SET-I, SET-II and SET-III, respectively. LA reduction (mmol/L) was greater after SET-III (estimate 1.8; 95% CI 1.0, 2.7) compared to SET-I. 

According to the AIC, age of the horse and SET had an effect on the HR_LA4_, whereas HR_LA2_ was only affected by age of the horse. HR_LA4_ (bpm) was higher in SET-II (estimate 15; 95% CI 7.1, 21.8) compared to SET-I, but did not differ in SET-III. Four-year old horses had lower HR_LA4_ (estimate −24; 95% CI −40.6, −7.0) and HR_LA2_ (estimate −19; 95% CI −31.9, −4.3) in all SETs compared to three-year-old stallions.

## 4. Discussion

This study demonstrated a negative effect on fitness progress of the training program used in 2020 where horses were on average trained for 144 ± 21 min divided over 4.7 ± 0.7 training sessions per week during the 70-day test in young Friesian stallions. The external workload of the three SETs was identical; however, physical effort increased in response to the same workload of the SET. This was shown by increasing HR and LA in SET-II and SET-III compared to SET-I. LA also increased between SET-II and SET-III. The number of horses used in the study is relatively low, and only four horses remained in the study until SET-III. However, regarding the selection process of approving stallions for a Dutch studbook, this is a normal and representative number of stallions that is getting approved. The statistical results and the performance of these four individual stallions showed that they had similar increasing HR and LA results as the other horses in each SET, demonstrating that these four horses were not outliers from the start. 

Repeated SETs are of value in training progress assessment in horses [14,32]. Ridden SETs are often used in riding horses since this is most representative of the normal workload of these horse. However, using a treadmill better standardizes the given exercise. In the present study, the rider was asked to maintain the same speed as much as possible in walk, trot and canter. A stopwatch was used to help maintain speed per round in the arena, and to have exact timing of the steps in the SETs [32,39]. The effects of the training program on SET performance were seen after 6 and 10 weeks of training. The decreased performance in the SETs after several weeks of training in the Friesian stallions could be explained by a status of overreaching or overtraining [2,54,55]. Overtraining is a complex syndrome, and the distinction between non-functional overreaching and overtraining is very difficult to make [5]. Overtraining occurs after a persisting imbalance between load and recovery [4,5,54]. Besides fatigue and decreased performance, behavioral changes such as mood changes and loss of appetite, changes in endocrine function and weight loss are described in relation to overtraining in humans and horses [5,13,54]. Several equine studies induced a state of overtraining by intense training for several weeks, leading to decreased fitness levels [13,55,56]. In these studies, horses performed less than before at standardized exercise tests, where HR and LA were increased [2,54,55]. In contrast, some equine and human studies show a decrease in lactate levels in overtrained athletes, but these studies used maximal exercise capacity for their SETs [5,56]. In the present study, using submaximal exercise tests, it seems that increased HR and LA measured during standardized exercise covering a 10-week training period indicates a reduction in fitness of these horses, since workload during the three SETs was identical. Body weight and behavior of the stallions were not assessed objectively.

The Friesian stallions were trained in the same frequency per week during the 70-day test as in the preparation period; however, training sessions in the 70-day test had a longer duration. External training load increased with 18% from the preparation period to the start of the 70-day test. This increase in external load combined withtraining 4.7 times per week, might have interfered with adequate recovery time. An inappropriate balance between training load and recovery time occurred for a period of 10 weeks. In the present study, five horses (31%) were eliminated because of lameness. Studies in human and equine athletes show a relation between the amount of increase in training load and injury risk [11,57,58,59,60]. In human athletes, a balanced workload reduces injury risks [11]. Regular intense exercise that is slowly increased with less than 10% increase per week leads to physical changes that protect against injuries in human athletes [3,9,10,11]. Studies in Thoroughbreds, Eventing horses and police horses show that fast changes in workload increase the risk of injury [24,61,62,63]. Inappropriately low or high training loads are associated with reduced performance in racehorses [21], and training loads that are within the range required for adaptation reduce injury risk [61]. All horses in the present study that were eliminated due to injury/lameness were eliminated in the first 6 weeks of the 70-day test. It is possible that the fast increase in workload from the preparation period to the 70-day test led to enlarged injury risks, which led, in turn, to the high number of horses being eliminated.

During the 70-day test, duration and frequency of the training sessions were comparable in the periods between SET-I and SET-II and the period between SET-II and SET-III, but fitness parameters in SET-III declined further compared to SET-II. Although the overall duration of the training did not change between SETs, the type of training changed between SET-II and SET-III as stallions were more trained for driving in the last period compared to the first period of 70-day test. The internal workload of driving compared to riding was not analyzed, but driving might have a different internal workload than riding, giving extra training load to the horses.

Compared to the young Friesian horses in studies of de Bruijn et al. and Munsters et al., the Friesian stallions in the present study seemed to be more fit at the start of the study, with lower HR and LA results after canter in SET-I [48,49]. However, in contrast to the present study, De Bruijn et al. found a positive effect of training within 8 weeks of training and a training frequency of 3 times a week. One of the SETs used was similar to the SET in the present study. In the study of de Bruijn et al., HR after canter-2 in the SET did not differ between SET-I and SET-II, but LA reduced from 5.0 ± 3.2 mmol/L to 3.0 ± 1.2 mmol/L after 8 weeks of training. Munsters et al. studied the fitness of 66 young Friesian mares in 2 SETs with 4 weeks of training in between. HR after canter-1 decreased between SET-I (177 ± 13 bpm) and SET-II (171 ± 12 bpm). Peak LA was not measured in SET-I but in SET-II it was 5.3 ± 2.0 mmol/L. The training protocol was not described in their study, but from author records it was shown to be approximately 5 times a week. The training protocol used in the study of de Bruijn et al. had less training sessions per week, although training load per training session was not measured, training three times a week seems to give enough time for recovery and thus allows fitness improvement. 

Transition to anaerobic metabolism is often defined as plasma lactate concentration >4 mmol/L, or onset of blood lactate accumulation (OBLA) during exercise. Previous studies have shown that young Friesian horses transfer to anaerobic metabolism, measured by LA > 4 mmol/L, on relatively low intensity work [48,49]. Using a single cut-off point for LA related to HR or velocity, such as HR_LA4_ or V_LA4_, is not the best predictor for the anaerobic threshold [64,65]; horses often cannot maintain LA constant for a longer period of time on that specific velocity or HR. A fixed cut-off value is often an overestimation of the lactate threshold [65,66,67]. Several tests are described to assess the intensity of exercise that leads to a steady state in lactate production, removal and metabolism and the threshold where anaerobic glycolysis participation starts to increase. The maximum lactate steady state (MLSS) is the gold standard to assess aerobic capacity in human athletes and horses [66,67]. In humans, the MLSS is defined as the highest constant workload/velocity that can be maintained for at least 25–40 min before anaerobic glycolysis participation starts to increase. To determine the MLSS, four to six exercise sessions at different speeds on different days are required. MLSS is not an easy measure to implement in training schedules of equine athletes and requires maximal exercise of the athlete. The lactate minimum speed test (LMS) can be performed in a single exercise test and gives an estimation of MLSS in human athletes. It can assess the individual blood lactate steady state without performance until exhaustion [64,65,66,67]. These tests can also monitor training effects over time. In the present study, a single cut-off as anaerobic threshold was used. Authors are aware that the results possibly overestimate the true anaerobic threshold.

While calculated parameters HR_LA4_ and HR_LA2_ increased from SET-I to SET-II, these values were lower in 4-year old horses compared to 3-year old horses. Besides exercise, HR is influenced by other factors such as environmental temperature, surface quality and stress. As temperature and surfaces were comparable between SETs, it seems logical that the increase in HR_LA4_ and HR_LA2_, as seen in this study, may be partly caused by an increase in stress levels of the horses. Older horses are more experienced with ridden work and might, for this reason, experience less tension, resulting in lower HR_LA4_ and HR_LA2_ values. 

On the selection days, during the preparation period, none of the horses reached LA values >4 mmol/L in contrast to the SETs during the 70-days test. The training the horses performed in their home environment seems appropriate for the exercise that was asked on the selection days. On the selection days, duration of canter was slightly shorter than canter performed during the SETs (two times 1.5 min compared to two times 2 min), and the period of walk between both canter bouts was longer on the selection days. De Bruijn et al. previously showed that low intensity breaks between canter bouts and two times 1 min canter instead of one time 2 min of canter lead to lower LA [49]. This explains the fact that the stallions had lower LA results during the selection days compared to SET-I in the 70-day test. During the 70-day test, the horses performed worse each SET, indicating an influence of the training program on performance during the SETs. 

External workload in the 70-day test was assessed using an IMU sensor to measure duration and time spent in different gaits. Using a sensor gives very precise data on external workload parameters [53]. In the present study, however, data from 59 out of a total of 491 training sessions were lost due to technical issues. A calculated replacement for these training sessions was used for further data analysis, and this might have caused a partial bias in the data. The most common reasons for missing workload data were problems with the phone application used to start and stop measurements. However, regarding workload calculations, leaving the days empty (no workload for that specific day) on the days where technical issues occurred would have given an even more incorrect representation of the actual situation as horses were trained that day. Authors are aware of the partial bias that is created due to the data replacement. Replacing data loss with horse and training specific averages seemed the most correct solution to this issue as this is seen as well in human studies regarding workload assessment. The data on training duration in the preparation period were obtained by questionnaires, which is less reliable than measurements using GPS or motion sensors [17]. Duration of training that trainers report and time that the horse was actually trained may differ, and this may have led to inaccuracy in our data. However, it was practically impossible to actually measure HR and duration of all 56 stallions during the preparation period as, at the beginning of the study, it was not known which stallions were going to be selected for the final 70-day test. Heart rate calculations or iTRIMP are useful estimates to evaluate internal workload in equine athletes [24,33,34]. Quantifying internal workload by measuring HR during all training sessions instead of only during SETs would have given more information on the impact of each training session on the horse; however, this was not possible due to technological limitations. Estimating internal workload of driving by measuring HR and LA on the horses would be a valuable addition to understand the risk of overtraining in the individual stallions. Additionally, De Meeûs d’Argenteuil et al. [5] showed changes in muscle diameters, fiber type and metabolic profiles after low intensity training in young Friesian horses. Measuring these muscle related parameters would provide additional information about muscular response to more intense training and would be valuable to consider in future studies. In the present study, however, the authors were limited in the samples that could be taken from the horses since these were elite privately owned horses in the run to become an approved studbook stallion. 

In the Netherlands, young Dutch Warmblood stallions undergo a similar performance test as part of their selection as future breeding stallions for the Royal Dutch Warmblood Studbook (KWPN). In these stallions, improved fitness was seen after 100-days of training using a lunging SET [47]. The Warmblood stallions had lower plasma lactate concentration (1.6 ± 0.7 mmol/L) after their first SET compared to the Friesian stallions in the present study. This can be explained by a lower intensity of the SET (lunging vs. ridden SET); however, the experienced effort for the same SET might be less for Warmblood horses than Friesian horses. The study of Harris et al. [31] even showed a higher workload while lunging compared to being ridden in Warmblood horses. The positive effect of this 100-day test might be explained by the fact that performing the same exercise seems to cause less effort in Warmblood sport horses compared to Friesian horses. The impact of workload on fitness progress of the 70-day test might therefore be breed-specific and cannot be directly translated to Warmblood horses. Interestingly, in the present study, 4-year-old horses were trained shorter than 3-year-old horses in the 70-day test. Dutch trainers of Friesian stallions usually start schooling their horses at the age of 2.5 years. The authors suspect that 4-year-old horses were further educated, and that riders and trainers were satisfied with the performance of the stallions sooner than with the younger horses that had less education in their shorter time of training. However, 4-year old horses did not seem to be more fit than 3-year old horses as HR and LA values during SETs did not differ between age groups. 

## 5. Conclusions

The current training program of the 70-day studbook approval test for young Friesian horses, with 144 min of training per week divided over 4.7 training sessions per week, leads to decreased fitness, suggesting non-functional overreaching or overtraining. This training program needs optimization, which could be achieved by reducing training frequency and duration. 

## Figures and Tables

**Figure 1 animals-13-00689-f001:**
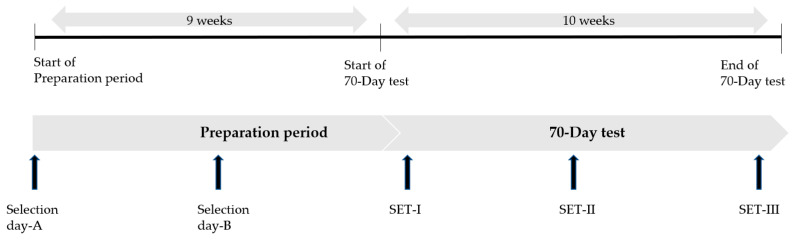
Training schedule of the young Friesian horses during the preparation period and the 70-day test, including standardized exercise tests (SET).

**Figure 2 animals-13-00689-f002:**
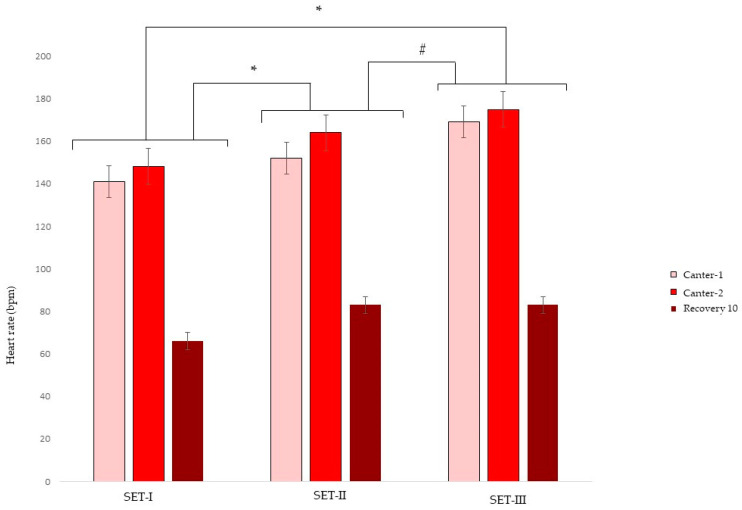
Mean ± SD heart rate (bpm) during three consecutive standardized exercise tests (SET-I, -II and -III) in young Friesian stallions with 5 (SET-I -SET-II) and 4 (SET-II–SET-III) weeks in between. * Indicating an overall statistical difference from SET-I; # indicating an overall statistical difference from SET-II.

**Figure 3 animals-13-00689-f003:**
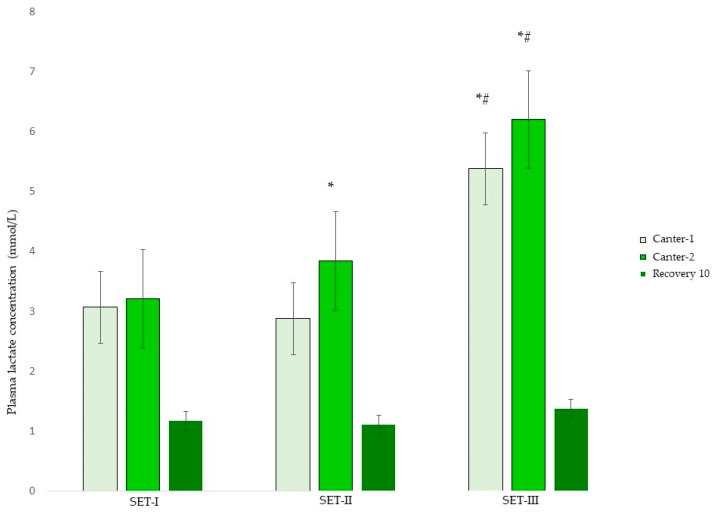
Mean ± SD plasma lactate concentration (LA) during three consecutive standardized exercise tests (SET-I, -II and -III) in young Friesian stallions with 5 (SET-I -SET-II) and 4 (SET-II–SET-III) weeks in between. * Indicating a statistical difference in LA per gait from SET-I; # indicating a statistical difference in LA per gait from SET-II.

**Table 1 animals-13-00689-t001:** Protocol of standardized exercise tests (SET) carried out in an in-door arena (20 × 60 m) by young Friesian horses, measuring heart rate (HR) and plasma lactate concentration (LA).

Time Table (Minutes)	Exercise	Plasma Lactate Sampling	Speed
SET	Indoor Arena under saddle		
00:00–01.00	Trot		
01:00–05:00	Walk		~2 m/s
05:00–07:00	Left trot		~3.5 m/s
07:00–09:00	Right trot		~3.5 m/s
09:00–10:00		LA	
10:00–12:00	Canter-1		~5.0 m/s
12:00–13:00		LA	
13:00–15:00	Canter-2		~5.0 m/s
15:00–16:00		LA	
16:00–26:00	Walk		Recovery ~2 m/s
26:00–27:00		LA	

**Table 2 animals-13-00689-t002:** Mean ± SD duration (min), plasma lactate concentration (LA, mmol/L) and heart rate (HR, bpm) at selection day-A and selection day-B in the preparation period before the 70-day test in young Friesian stallions.

	Stallions Not Accepted in 70-Day Test		Stallions Accepted in 70-Day Test	
	Selection Day-A (*n* = 25)	Selection Day-B (*n* = 21)	Selection Day-A (*n* = 11)	Selection Day-B (*n* = 11)
Duration total (min)	31 ± 9	29 ± 5	29 ± 6	23 ± 6
Duration warm-up (min)	16 ± 7	13 ± 5	15 ± 4	10 ± 6
Duration test (min)	11 ± 1	11 ± 1	10 ± 1	10 ± 1
Mean HR total (bpm)	99 ± 8	100 ± 9	106 ± 8	103 ± 10
Mean HR test (bpm)	117 ± 9	116 ± 11	122 ± 10	116 ± 7
LA after test (mmol/L)	1.8 ± 0.6	1.8 ± 0.8	2.0 ± 0.8	1.8 ± 0.4
LA recovery (mmol/L)	1.6 ± 0.3	1.8 ± 0.4	1.6 ± 0.4	1.8 ± 0.2
Horses LA > 2.0 (*n*)	11	11	3	4

**Table 3 animals-13-00689-t003:** External workload (duration and number of training sessions) per type, gait and per week of young Friesian stallions during the preparation period (*n*= 9 weeks) and the 70-day test. NA = not available.

	Preparation Period	70-Day Test	
		Period between SET-I and II	Period between SET-II and III
Number of training sessions per week			
Total	4.5 ± 0.9	4.7 ± 0.7	4.7 ± 1.0
Dressage	3.1 ± 0.7	3.5 ± 0.5	2.2 ± 1.8
Lunging	1.3 ± 0.4	0.1 ± 0	0.3 ± 0.6
Driving	0.5 ± 0.11	0.7 ± 0	2.0 ± 0.5
Duration (min/week)			
Total	122 ± 25	144 ± 21	131 ± 39
Walk	NA	94 ± 22	82 ± 28
Trot	NA	37 ± 11	41 ± 10
Canter	NA	10 ± 3	8 ± 6

## Data Availability

The data presented in this study are available on request from the corresponding authors.

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
