# Peer review of "Longitudinal Training and Workload Assessment in Young Friesian Stallions in Relation to Fitness: Part 1"

_animals, 2023, doi:10.3390/ani13040689_

Round 1

Reviewer 1 Report

The main question addressed by the research is the workload during training of Friesian stallions. This topic is relevant in the field for trainer of the horses because the horses should not be overtrained. This is very relevant for animal welfare!

This is the first time using Frisian stallions in training and measure the workload. The authors did the correct methodology, the conclusion is short and correct, the references appropriate, and the tables are understandable and clear.

However, the trainers do not know the highest tolerable heart rate. If they use the table 3, they would not overtrain the horse. This is the correct animal welfare.

Other comments:

Dear Authors, please change the word "psychological" to "behavioural" on page 2 because the horse is an animal and not a human being!

Author Response

Dear reviewer, thank you for reviewing our manscript and for your comments. We have tried to address them as good as possible. Please see our answers below in red.

The main question addressed by the research is the workload during training of Friesian stallions. This topic is relevant in the field for trainer of the horses because the horses should not be overtrained. This is very relevant for animal welfare!

This is the first time using Frisian stallions in training and measure the workload. The authors did the correct methodology, the conclusion is short and correct, the references appropriate, and the tables are understandable and clear.

However, the trainers do not know the highest tolerable heart rate. If they use the table 3, they would not overtrain the horse. This is the correct animal welfare.

We thank the reviewer for pointing this out. Table 3 in the original manuscript (table 2 in the revised manuscript) represents the results from the selection days in the Preparation period. In this Preparation period horses are trained in their normal environment and on the selection days they have to perform a dressage test for judges. We agree with the reviewer that, based on the findings of the selection days, horses perform well and the impact of this dressage test is not too high for the horses. In other words, the horses have acceptable HR and LA results during and after the test, indicating that the workload is appropriate for the horse, and horses never reach a plasma lactate > 4 mmol/L. Thus in the Preparation period the workload that the horses perform in their training in their home environment prepares them well for the Selection Days.  This finding is clearly different from the results obtained during SET-I, II and III that horses performed during the 70-Day test following the Preparation period, where horses have higher HR and LA results, that increase in SET-II and SET-III. We comment on this in the discussion with the following paragraph:

“On the selection days, during the Preparation period, none of the horses reached LA values >4 mmol/L in contrast to the SETs during the 70-Days test. The training the horses performed in their home environment seems appropriate for the exercise that was asked on the selection days. On the selection days, duration of canter was slightly shorter than canter performed during the SETs (two times 1,5 min compared to two times 2 min), and the period of walk between both canter bouts was longer on the selection days. De Bruijn et al. previously showed that low intensity breaks between canter bouts and two times 1 min canter instead of one time 2 min of canter lead to lower LA [46]. This explains the fact that the stallions had lower LA results during the selection days compared to SET-I in the 70-Day test. During the 70-Day test the horses performed worse each SET, indicating an influence of the training programme on performance during the SETs. “

When reviewer and editor would like to see this addressed even more in the paper we are happy to add comments in the discussion.

Other comments:

Dear Authors, please change the word "psychological" to "behavioural" on page 2 because the horse is an animal and not a human being!

Thank you for your comment, this word has been changed as suggested

Reviewer 2 Report

With interest, I have been reading the manuscript entitled: "Longitudinal Training and Workload Assessment in Young 2 Friesian Stallions in Relation to Fitness: Part 1 " by Siegers et al.

Congratulations on the manuscript. I have imagined how much hard work was involved, and it may have contributed to other researchers' training within the research group. Overall, the article will contribute to understanding the longitudinal training in the Friesian stallions and is worthy of publication. The work is written in understandable language. My main concern with this manuscript is the need for more other information to characterize overtraining in these Friesian horses.

Suggestions for improving the manuscript:

Abstract:

My comments: Well written and straightforward.

Introduction

L 96-97: “Young Friesian horses seem to have a higher aerobic energy expenditure in trot and canter, shown by higher HR and LA results”

My comments: the authors should explain why the “higher aerobic energy” induces higher plasma lactate concentration (LA).

Materials and Methods

My comments: In my opinion the MM is very hard to follow, same with the table 1. I suggest that authors present a study workflow (schematic representation of the experimental protocol), including the SETs.

L 215: Typo: “ware” analyzed or were analyzed?

Results

Table 6:

My comments: This reviewer is not convinced that this table really helps much. I would like to see the symbols indicating statistical significance (maybe an asterisk?). Otherwise, it called my attention to the lactate concentration recovery rate in the SETs. Comparing the “Canter -2” vs “Recovery 10”, I have found a lactate recovery rate of 68.5, 73.4, and 78% in the SET-I, SET-II, and SET-III, respectively. Are they significant? Please clarify, elaborate, and discuss and review the current study findings. 

Discussion

Lines 377-378:Transition to anaerobic metabolism can be defined as plasma lactate concentration >4 mmol/L, or onset of blood lactate accumulation (OBLA) during exercise”

My comments: The authors could take into account the new approach for transition to anaerobic metabolism in horses. In horses, there is increasing evidence that a fixed cut-off value for plasma lactate, as 4 mM - OBLA, is not a solid predictor for the lactate threshold (aerobic/anaerobic transition). For more details, see:

DOI: 10.3389/fphys.2022.792052

DOI: 10.1080/00480169.2013.815103

DOI: 10.2527/jas.2009-2693

DOI: 10.1590/S0102-09352014000100007

How would the authors cover a lot of ground in the studies mentioned above over their results? Please clarify and elaborate more when discussing and reviewing the current study findings. 

Overtraining issues:

My comments: This reviewer is not convinced that the reduced performance in SETs represents the “status of overreaching or overtraining”. The authors have suggested in lines 329, 423, and 447 overtraining syndrome (OTS) considering the rise in heart rate and lactate results. OTS reflects the complex essence of the condition, the multifactorial aetiology, and that an imbalance/inequality between training loading and recovery may be the primary reason. Poor nutrition, mood changes, weight loss, and loss of appetite are all variously reported components of the OTS condition. Have you measured anything else that can strengthen the OTS suggestion? If not, consider putting this as a study limitation. Please clarify, elaborate, and discuss and review the current study findings. 

Author Response

With interest, I have been reading the manuscript entitled: "Longitudinal Training and Workload Assessment in Young 2 Friesian Stallions in Relation to Fitness: Part 1 " by Siegers et al.

Congratulations on the manuscript. I have imagined how much hard work was involved, and it may have contributed to other researchers' training within the research group. Overall, the article will contribute to understanding the longitudinal training in the Friesian stallions and is worthy of publication. The work is written in understandable language. My main concern with this manuscript is the need for more other information to characterize overtraining in these Friesian horses.

Dear reviewer, 

Thank you for your comments on our manuscript, we have tried to address them satisfactory. Please find our replies below in red.

Suggestions for improving the manuscript:

Abstract:

My comments: Well written and straightforward.

Introduction

L 96-97: “Young Friesian horses seem to have a higher aerobic energy expenditure in trot and canter, shown by higher HR and LA results”

My comments: the authors should explain why the “higher aerobic energy” induces higher plasma lactate concentration (LA). Thank you for this suggestion. We have now changed that part of the introduction to: “Young Friesian horses seem to have a higher aerobic energy expenditure in trot and canter. They show higher HR and LA results in comparison to Warmblood horses with comparable backgrounds, trotting and cantering at the same speed [44,45,46], indicating that the transition to anaerobic metabolism starts at lower exercise speeds in Friesian horses.”

Materials and Methods

My comments: In my opinion the MM is very hard to follow, same with the table 1. I suggest that authors present a study workflow (schematic representation of the experimental protocol), including the SETs.

We thank the reviewer for pointing this out, we have now replaced table 1 with a workflow/schematic timeline (figure 1) to improve readability

L 215: Typo: “ware” analyzed or were analyzed? This typo has been changed to “were”

Results

Table 6:

My comments: This reviewer is not convinced that this table really helps much. I would like to see the symbols indicating statistical significance (maybe an asterisk?). We thank the reviewer for this suggestion. Table 6 has now been replaced with figure 2 and 3, two graphs with the HR and LA results, and an indication of statistical differences between SETs using an asterisk.

Otherwise, it called my attention to the lactate concentration recovery rate in the SETs. Comparing the “Canter -2” vs “Recovery 10”, I have found a lactate recovery rate of 68.5, 73.4, and 78% in the SET-I, SET-II, and SET-III, respectively. Are they significant? Please clarify, elaborate, and discuss and review the current study findings. We thank the reviewer for this suggestion and added LA recovery rates in the LA results section. To clarify further, in the statistical model that was used in the study, recovery plasma lactate was included. Recovery 10 LA was lower than LA in canter-1 and canter-2, however there was no difference in recovery 10 LA between the different SETs according to the statistical model. The maximum LA is higher in SET-III than in SET-I or SET-II, whereas recovery LA did not change significantly in the different SETs. The recovery rate, or change in LA was higher  in the final SET compared to SET-I. We have now added the recovery rates in the LA results.

Discussion

Lines 377-378: “Transition to anaerobic metabolism can be defined as plasma lactate concentration >4 mmol/L, or onset of blood lactate accumulation (OBLA) during exercise”

My comments: The authors could take into account the new approach for transition to anaerobic metabolism in horses. In horses, there is increasing evidence that a fixed cut-off value for plasma lactate, as 4 mM - OBLA, is not a solid predictor for the lactate threshold (aerobic/anaerobic transition). For more details, see:

DOI: 10.3389/fphys.2022.792052

DOI: 10.1080/00480169.2013.815103

DOI: 10.2527/jas.2009-2693

DOI: 10.1590/S0102-09352014000100007

How would the authors cover a lot of ground in the studies mentioned above over their results? Please clarify and elaborate more when discussing and reviewing the current study findings. Thank you for this comment, the mentioned literature and discussion related to the present study is now included in the paragraph about the anaerobic threshold.

Overtraining issues:

My comments: This reviewer is not convinced that the reduced performance in SETs represents the “status of overreaching or overtraining”. The authors have suggested in lines 329, 423, and 447 overtraining syndrome (OTS) considering the rise in heart rate and lactate results. OTS reflects the complex essence of the condition, the multifactorial aetiology, and that an imbalance/inequality between training loading and recovery may be the primary reason. Poor nutrition, mood changes, weight loss, and loss of appetite are all variously reported components of the OTS condition. Have you measured anything else that can strengthen the OTS suggestion? If not, consider putting this as a study limitation. Please clarify, elaborate, and discuss and review the current study findings. 

Thank you for your comments, we have not measured behavioural parameters or body weight in the present study. We have now added this comment to the discussion about overtraining, and added information about the complexity of defining overtraining. We agree that additional findings such as mood changes and weight loss or reduced appetite would have strengthened our suspicion of overtraining. However, we still consider overtraining as an important explanation for the reduced performance in the repeated SETs.

Reviewer 3 Report

Dear authors, the topic of your work “Longitudinal Training and Workload Assessment in Young 

Friesian Stallions in Relation to Fitness: Part 1 

” it is very scientific importance in equine exercise physiology and functional selection.

I would like the following considerations to the authors:

Simple summary and Abstract are corrected summarize. 

1. Introduction 

The introduction is fine except that it would be necessary to include if there are any previous studies on exercise physiology and response to training in Friesian horses, information on muscle fibre type, enzyme endowments, etc. It would be very convenient to include it in this article as it is a very good work that you present. 

2. Materials and Methods 

Line 126:  2.1.1. Preparation Period and Selection Days. Are the selection criteria on exterior traits and gaits, subjective based in their experience? If is like this, please clarify. 

The Study deign, is well planned. My recommendation is that for future studies you should consider using a treadmill to standardise the conditions of speed, time and slope of the exercise loads.

3.- Results and discussion. 

3.2.2. Standardised Exercise Tests (SETs) 

I advise the authors to prepare some graphs with the data in Table 6 in order to compare in a more visual way the evolution of the parameters studied during the different phases of training.

The explanation of the results in table 6, explained in lines 300-312, should be set out more clearly, in some graph or diagram, or even table with the interactions discussed.

4.- Conclusions. 

When talking about overtraining, no blood analysis of muscle bio-indicators is provided, and if they had such data, it would be interesting if they could provide them to reinforce the conclusions of this work.

Author Response

Dear authors, the topic of your work “Longitudinal Training and Workload Assessment in Young 

Friesian Stallions in Relation to Fitness: Part 1 

” it is very scientific importance in equine exercise physiology and functional selection.

Dear reviewer,

Thank your for your comments on our manuscript. We have tried to address your suggestions as good as possible. We have answered on your comments in red. 

I would like the following considerations to the authors:

Simple summary and Abstract are corrected summarize. 

  1. Introduction 

The introduction is fine except that it would be necessary to include if there are any previous studies on exercise physiology and response to training in Friesian horses, information on muscle fibre type, enzyme endowments, etc. It would be very convenient to include it in this article as it is a very good work that you present. We thank the reviewer for this comment and the findings of this study are now added to the introduction: “De Meeûs d’Argenteuil et al. performed a study in untrained Friesian horses undergoing an 8-week low intensity training protocol on a treadmill or in an aquatrainer. The increase in muscle diameter was the most after 4 weeks of training. Training intensity did not change after 4 weeks of training and muscle diameters decreased again. These results suggest that in an appropriate training protocol a slight increase in workload should be made after 4 weeks of training. Shifts in muscle fiber type and metabolic profiles occured after the training protocols, with different muscles showing different changes [47].”

  1. Materials and Methods 

Line 126:  2.1.1. Preparation Period and Selection Days. Are the selection criteria on exterior traits and gaits, subjective based in their experience? If is like this, please clarify. We thank the reviewer for this comment and tried to elaborate this more deeply: The selection criteria are subjective, and therefore the word “subjective” is now added to the selection criteria by the judges.

The Study deign, is well planned. My recommendation is that for future studies you should consider using a treadmill to standardise the conditions of speed, time and slope of the exercise loads. Thank you for this suggestion. The authors agree that using a treadmill would improve standardization of the SET’s, however in this practical study it was not possible to use a treadmill for the SET as horses’ owners would not give us permission to train and test their horses on a treadmill. In addition, we  wanted to evaluate the horses under saddle, since this is the most representative for the normal workload that is given to the horses. We added the next sentence to the discussion to comment on using a treadmill:

“Ridden SETs are often used in riding horses, since this is most representative for the normal workload of these horse. However, using a treadmill standardizes the given exercise more. In the present study, the rider was asked to maintain the same speed as much as possible in walk, trot and canter. A stopwatch was used to measure help maintaining speed per round in the arena, and to have exact timing of the steps in the SETs [29,36].

3.- Results and discussion. 

3.2.2. Standardised Exercise Tests (SETs) 

I advise the authors to prepare some graphs with the data in Table 6 in order to compare in a more visual way the evolution of the parameters studied during the different phases of training.

The explanation of the results in table 6, explained in lines 300-312, should be set out more clearly, in some graph or diagram, or even table with the interactions discussed. We thank the reviewer for this comment. Table 6 has now been replaced with figure 2 and 3, two graphs with the HR and LA results, and indication of statistical differences between SETs.

4.- Conclusions. 

When talking about overtraining, no blood analysis of muscle bio-indicators is provided, and if they had such data, it would be interesting if they could provide them to reinforce the conclusions of this work.

Thank you for this suggestion. Since we were working with privately owned horses being in the running to become an approve studbook stallion we were limited in samples we could obtain from the horses. We do not have further blood analysis than plasma lactate concentration, nor were we able to take muscle biopsies of these privately owned stallions. We have now added a sentence in the discussion that this information would be valuable to add when practically possible. “Additionally, De Meeûs d’Argenteuil et al. [47] showed changes in muscle diameters, fiber type and metabolic profiles after low intensity training in young Friesian horses. Measuring these muscle related parameters would provide additional information about muscular response to more intense training and would be valuable to consider in future studies. In the present study however the authors were limited in the samples that could be taken from the horses since these were elite privately owned horses in the run to become an approved studbook stallion.”